# A Proof-of-Concept Study Using Numerical Simulations of an Acoustic Spheroid-on-a-Chip Platform for Improving 3D Cell Culture

**DOI:** 10.3390/s21165529

**Published:** 2021-08-17

**Authors:** Arash Yahyazadeh Shourabi, Roozbeh Salajeghe, Maryam Barisam, Navid Kashaninejad

**Affiliations:** 1Department of Mechanical Engineering, Sharif University of Technology, Tehran 11155, Iran; arash.knt.1374@gmail.com (A.Y.S.); roozbehsalajeghe@gmail.com (R.S.); barisam_m@mech.sharif.ir (M.B.); 2Queensland Micro- and Nanotechnology Centre, Nathan Campus, Griffith University, 170 Kessels Road, Brisbane, QLD 4111, Australia

**Keywords:** lab-on-chip, acoustic microfluidics, spheroid-on-chip, necrotic, quiescent zones

## Abstract

Microfluidic lab-on-chip devices are widely being developed for chemical and biological studies. One of the most commonly used types of these chips is perfusion microwells for culturing multicellular spheroids. The main challenge in such systems is the formation of substantial necrotic and quiescent zones within the cultured spheroids. Herein, we propose a novel acoustofluidic integrated platform to tackle this bottleneck problem. It will be shown numerically that such an approach is a potential candidate to be implemented to enhance cell viability and shrinks necrotic and quiescent zones without the need to increase the flow rate, leading to a significant reduction in costly reagents’ consumption in conventional spheroid-on-a-chip platforms. Proof-of-concept, designing procedures and numerical simulation are discussed in detail. Additionally, the effects of acoustic and hydrodynamic parameters on the cultured cells are investigated. The results show that by increasing acoustic boundary displacement amplitude ($d_{0}$), the spheroid’s proliferating zone enlarges greatly. Moreover, it is shown that by implementing $d_{0}\;$ = 0.5 nm, the required flow rate to maintain the necrotic zone below 13% will be decreased 12 times compared to non-acoustic chips.

## 1. Introduction

Cancer is the second leading cause of death globally, accounting for more than 8 million deaths per year [1]. One of the first crucial steps in the battle against this disease is understanding the underlying processes happening during tumorigenesis and finding the different affecting factors [2]. A popular method to mimic the tumor’s microenvironment and study its behaviors is to employ tumor spheroids [3]. Spheroids are three-dimensional (3D) cellular masses formed from the aggregation of cells under special culturing conditions [4]. The morphology, growth, and cell interactions in spheroids are similar to those in actual tumors, making spheroid culture both an appropriate in vitro model of tumor and a valuable tool for testing drugs and treatment efficiency [5,6].

For many years, methods such as hanging drops and 96-well plates have been used to form and study tumor spheroids [7]. However, the static conditions in these methods lead to accelerated depletion of nutrients and accumulation of waste, adversely affecting spheroid growth and leading to false results on drug candidates [8].

On the other hand, microfluidic platforms have long been used for various sensing and actuating applications [9,10]. Recently, these devices have been developed to better simulate the continuous perfusion conditions of actual tumors [11,12,13,14,15]. Such systems are increasingly being used to mimic the different key phenomena of the tumor micro-environment, such as aggregation, formation, growth, angiogenesis, and metastasis of cancer tumor cells [15,16]. In addition, hydro-dynamically trapping the cells in these devices leads to highly uniform size distributions of spheroids [17], since the fluidic conditions in microchannels are precisely adjustable and feasibly controllable [18]. One of the most common techniques used to generate and culture these spheroids on microfluidic chips is the integration of microchannels and microwells [15,19].

The cell suspension is first introduced at the inlet for the spheroid generation, reaching the microwells through interconnected microchannels [20]. Then, the trapped cells inside these microwells start to aggregate and subsequently a multicellular spheroid structure is formed by self-assembling of cells [21]. The continuous, long-term flow of culture media through the channels then provides the cells with their required nutrition and removes cells’ waste from the system [22].

Depending on the availability of oxygen and nutrients, there are two critical regions in a tumor spheroid, namely necrotic and quiescent regions [23,24]. Massive lack of oxygen and nutrients in a necrotic region leads to cell death [25]. A quiescent region is where the lack of oxygen and nutrients causes the cells to secrete growth factors that may lead to angiogenesis [22,26,27]. Barisam et al. numerically studied the necrotic cores and quiescent zones in spheroids cultured in U-shaped barrier microfluidic chips and investigated the effect of several hydrodynamic parameters on multicellular aggregates in such platforms [26]. In another work, toroidal and spherical 3D cellular aggregates were simulated numerically in both microwells and U-shaped barrier chips [22]. It was shown that although the U-barrier design provides spheroids with a better concentration of oxygen, it also exposes cells to higher values of fluid shear stress, which is a negative point in culturing spheroid on microchips. Grimes et al. investigated the effect of spheroid diameter on necrotic and hypoxic development both numerically and experimentally and concluded that an increase in diameter would decrease the percentage of the proliferating zone [28]. Recently, Im et al. studied the hypoxic zone of stem cell spheroids and their gene expression experimentally and discussed how the seeding density affects the compaction of the spheroid and the hypoxia microenvironemnt [29].

Consequently, based on the literature, changing the microchip’s geometry, increasing the media’s flow rate, and decreasing spheroids’ diameters are among the conventionally proposed solutions to battle the growth of necrotic and quiescent zones [15]. Changing the geometry is not always a good solution because of the difficulties in the fabrication of other designs and challenges corresponding to their operation and function. These challenges can induce high shear stresses on cells and make initial cell seeding even more arduous [15,22,26]. Moreover, culturing small spheroids is not always favorable since they may not resemble the in vivo geometry of cancerous tumors.

Increasing the flow rate is another possible solution to prevent necrotic cores of in vitro spheroid culture platforms. While this approach improves the nutrition distribution inside the spheroid, it suffers from serious downsides: (1) Higher flow rates require more culture media and drugs in long-term culturing and drug screening. This increment in reagent consumption might be too expensive in some cases, and it is in contrast with the goal of using microfluidic technology, which is less consumption of materials and reagents [30]. (2) A higher value of fluid shear stress will be imposed on the spheroid, which can damage the plasma membranes of the cells forming the spheroid and may lead to cellular damage and death [31]. (3) By increasing the flow rate, the lift force on the spheroid rises and the spheroid may move to the outlet and become wasted [32].

On the other hand, acoustofluidics is an emerging field that can address the potential problems associated with conventional microfluidic platforms. A comprehensive review of the application of bulk and surface acoustic waves in microfluidics has been conducted in the acoustofluidics tutorial series [33] and other reviews with a principal focus on surface acoustic waves [34,35]. Many studies have targeted microfluidic acoustic waves for biological purposes, especially for cell/particle manipulations. For example, Li et al. have utilized surface acoustic waves to separate circulating tumor cells from blood samples [36]. Using concentrated surface acoustic waves, Shilton et al. managed to aggregate and separate particles and mix fluids within a microfluidic droplet [37]. Incorporating standing surface acoustic waves into their microfluidic chip, Li et al. co-cultured cancer cells and endothelial cells in their novel study [38]. In a relevant work, Greco et al. managed to highly increase the cell-proliferation rate in a microfluidic chip with surface acoustic waves compared to its static condition [39]. Integration of spheroid culturing kits with acoustic waves is the core of some recent papers, but they mainly utilized acoustics for the cell seeding step and spheroid fabrication, and not for the idea of mass transport enhancement [40,41]. Chen et al. used acoustic waves on their microfluidic platform to fabricate spheroids [42]. In their work, the formation of hypoxic regions within spheroids was also investigated experimentally utilizing live/dead cell staining kits. Regarding the modeling of acoustofluidics, Muller et al. adopted perturbation theory to model the acoustic streaming phenomenon inside a microfluidic chip [43]. Raghavan et al. developed a numerical model to simulate the acoustic streaming effect inside a droplet. Although their model predicted the experimental velocity trends inside the droplet, it was not accurate, as the velocity across the height of the droplet obtained from simulations was an order of magnitude lower than the one obtained from experiments [44].

To address the aforementioned challenges in spheroid-on-chip microsystems, in this study we took a step forward towards integrating acoustic microfluidics with conventional spheroid-on-chip platforms as a novel technique to decrease consumption of the reagents and decrease the associated shear stress on the cultured cells. The proposed model suggests a new culturing approach that can shrink the necrotic and quiescent zones in spheroids while preventing spheroids eluding the well and avoiding intense fluid shear stresses in the system. The present concept is feasible for fabrication and operation and can also be the starting point of efficient, low-cost acoustic cell culture platforms.

The acoustic field inside a microchannel directly affects the flow pattern, thereby indirectly influencing the mass transport processes that are vital to living spheroids on the chip [45]. Here, we conduct a proof-of-concept study in the implementation of acoustic waves in microfluidic cell culture platforms and numerically show that such a platform leads to an improvement in the culturing conditions of spheroids and a reduction in the consumption of culture media in these devices. To do so, the numerical simulation of the system is discussed in detail and the effects of acoustic and hydrodynamics parameters on cultured cells are investigated. Moreover, a comparison between on-chip spheroid culturing with and without acoustofluidic integration is carried out to better illustrate the benefits of integrating acoustic fields into spheroid-on-chip systems.

## 2. Materials and Methods

### 2.1. Geometry and Model Description

The geometry of the system is illustrated in Figure 1. The spheroid is cultured in a microwell located beneath the perfusion microchannel. The dimensions of the microfluidic platform and the 3D cell aggregate are listed in Table 1. Culture media passes through the perfusion channel and leaves the system via the outlet. In this way, nutritious species are brought to the system continuously and are transported to the spheroid via diffusion. At the bottom surface of the microwell, an ultrasonic piezoelectric transducer is located, which vibrates at its resonance frequency and propagates acoustic waves to the flow when connected to electrical voltage. The boundary vibration disturbs the flow and brings about a more homogenous culture media through the acoustic streaming phenomenon.

### 2.2. Governing Equations

Based on the physics of the problem, a 2D computational domain has been considered for simulation 1. The culture medium’s properties are assumed to be similar to those of water at 37 °C [26]. Moreover, in this study, dissolved oxygen and glucose in the culture media are the main parameters of interest that will be discussed in detail.

#### 2.2.1. Microfluidic Flow

We consider an incompressible and steady laminar flow of culture medium inside the channel and around the spheroid. As such, the continuity and momentum equations can be written as follows [26]:(1)$$\vec{\nabla }\cdot \vec{V}=0$$
(2)$$\rho \left(\vec{\nabla }\cdot \vec{V}\right)\vec{V}=-\vec{\nabla }p+\mu \nabla^{2}\vec{\nabla }+\vec{F}$$
where $\vec{V}$ is velocity vector, *p* is the pressure of the fluid, and ρ and μ are density and viscosity, respectively. $\vec{F}$ represents the net value of the volume forces, which is due to the forces that the acoustic field exerts on the flow. The flow velocity inside the spheroid is set to be zero, and its outer surface works as an impermeable wall for the medium flow.

#### 2.2.2. Transport of Dilute Species

In order to model the spheroid, a continuum modeling approach was implemented. The other model, namely the discrete approach, is suitable for the simulation of cellular aggregates’ growth since it has control over cell interactions. The continuum approach is suitable for macroscopic analysis with a focus on epigenetic variations such as environment-related variables. As such, the general form of the diluted species mass transfer equation applies [26]:(3)$$\;\partial_{t}c_{i}+\vec{V}\cdot \vec{\nabla }c_{i}=D_{F.i}\vec{\nabla }c_{i}+R_{i}$$
where $c_{i}$ represents the concentration of oxygen/glucose, *t* is time, and $D_{F.i}$ is the respective diffusion coefficient of either oxygen or glucose. $R_{i}$ models oxygen/glucose consumption by living cells in the computational domain and is defined by the Michaelis-Menten reaction equation as follows [46]:(4)$$R_{i}=\frac{V_{max}c_{i}}{c_{i}+K_{m}}$$
where $V_{max}$ is the maximum reaction rate and $K_{m}$ is Michaelis-Menten constant.

To model the culture media, the steady-state form mass transfer without the effect of cells can be used. As such, the transport phenomenon of culture media flow is purely governed by diffusion and convection, and Equation (3) can be simplified as follows:(5)$$\vec{V}\cdot \vec{\nabla }c_{i}=D_{F.i}\vec{\nabla }c_{i}$$

Inside the spheroid, there are no flows. Only oxygen/glucose consumption by cells and diffusion of species within the spheroid’s tissue exist. Consequently, the convective term of Equation (3) should be omitted, and finally, Equation (6) controls the mass transport inside the spheroid’s inner domain:(6)$$0=D_{F.i}\vec{\nabla }c_{i}+R_{i}$$

#### 2.2.3. Acoustic

Considering a compressible fluid, the governing equations are as follows:(7a)$$\partial_{t}\rho =-\nabla \cdot \left(\rho V\right)$$
(7b)$$\rho \;\partial_{t}\vec{V}=-\vec{\nabla }p-\rho \left(\vec{V}\cdot \vec{\nabla }\right)\vec{V}+\mu \nabla^{2}\vec{V}+\beta \mu \vec{\nabla }\left(\nabla \cdot \vec{V}\right)$$

Equation (7a) is the well-known continuity equation for a compressible fluid, and Equation (7b) is the general Navier-Stokes equation by considering the transient (the term added in the left-hand side of the equation) and compressibility (added to the right-hand side of the equation, in which β stands for viscosity ratio) effects.

As this nonlinear equation does not have an analytical solution, perturbation theory is a conventional method used to obtain a reasonable approximate solution. In this method, a quiescent fluid with constant density $\rho_{0}$, constant pressure $p_{0}$, and zero velocity is considered before the wave incidence. As the wave influences the fluid, it disturbs the flow. Considering very small changes in the flow properties, the following first-order approximation can be made:(8a)$$\rho =\rho_{0}+\rho_{1}$$
(8b)$$p=p_{0}+p_{1}$$
(8c)$$T=T_{0}+T_{1}$$
(8d)$$V=0+V_{1}$$

Considering the process to be isentropic, pressure can be written as a function of density:(9)$$p\left(\rho \right)=p_{0}+{\left(\frac{\partial p}{\partial \rho }\right)}_{s}\rho_{1}=p_{0}+c_{a}^{2}\rho_{1}$$
in which $c_{a}$ is the isentropic speed of sound defined as $c_{a}=\sqrt{{\left(\frac{\partial p}{\partial \rho }\right)}_{s}}$. Substituting Equations (8) and (9) into Equation (7) and solving for the first-order terms results in:(10a)$$\partial_{t}\rho_{1}=-\rho_{0}\nabla \cdot V_{1}$$
(10b)$$\rho_{0}\partial_{t}V_{1}=-c_{a}^{2}\nabla \rho_{1}+\eta \nabla^{2}V_{1}+\beta \eta \nabla \left(\nabla \cdot V_{1}\right)$$
(10c)$$\partial_{t}T_{1}=D_{th}\nabla^{2}T_{1}+\frac{\alpha T_{0}c_{a}^{2}}{\rho_{0}C_{p}}\partial_{t}\rho_{1}$$

In obtaining Equation (10), the product of 1st order terms have been neglected as their value is small compared to the other ones.

Navier-Stokes equations are nonlinear in general, and approximating them by a first-order term may introduce some errors into the final results. As a result, it is common practice to continue Equation (8) to second-order terms (for the sake of simplicity, the second-order energy equation has been neglected [43]):(11a)$$\rho =\rho_{0}+\rho_{1}+\rho_{2}$$
(11b)$$p=p_{0}+p_{1}+p_{2}$$
(11c)$$V=0+V_{1}+V_{2}$$

Using the definition of the sound velocity, it is possible to write the pressure as a function of density as follows:(12)$$p=p_{0}+p_{1}+p_{2}=p_{0}+c_{a}^{2}p_{1}+0.5\left(\partial_{\rho }c_{a}^{2}\right)\rho_{1}^{2}$$

Inserting Equations (11) and (12) into the Navier-Stokes equations (Equation (7)) and solving for the second-order terms yields:(13a)$$\partial_{t}\rho_{2}=-\rho_{0}\nabla \cdot V_{2}-\nabla \cdot \left(\rho_{1}V_{1}\right)$$
(13b)$$\rho_{0}\partial_{t}V_{2}=-\nabla p_{2}+\eta \nabla^{2}V_{2}+\beta \eta \nabla \left(\nabla \cdot V_{2}\right)-\rho_{1}\partial_{t}V_{1}-\rho_{0}\left(V_{1}\cdot \nabla \right)V_{1}$$

In the derivation of Equation (13), the product of higher-order terms has also been neglected. Generally, second-order terms are negligible compared to the first-order ones, except for the harmonic-time-dependent cases in which the time average of the terms in Equation (10) disappears; however, some of the terms in Equation (13) have a non-zero time average [47]. Defining 〈F〉 as the time average function for the variable F(t):(14)$$\langle F\rangle \equiv \frac{1}{\tau }\overset{\tau }{\underset{0}{\int }}F\left(t\right)dt$$
and considering the following harmonic time-dependence for the acoustic variables (ω is the angular frequency and is equal to ω=2πf, in which f is the frequency):(15a)$$p_{i}\left(r,t\right)=p_{i}\left(r\right)e^{-i\omega t},\;i=1,2$$
(15b)$$\rho_{i}\left(r,t\right)=\rho_{i}\left(r\right)e^{-i\omega t},\;i=1,2$$
(15c)$$V_{i}\left(r,t\right)=V_{i}\left(r\right)e^{-i\omega t},\;i=1,2$$

Upon inserting these terms into Equation (13), this set of equations converts to the following ones:(16a)$$\rho_{0}\nabla \cdot \langle V_{2}\rangle =-\nabla \cdot \langle \rho_{1}V_{1}\rangle$$
(16b)$$\eta \nabla^{2}\langle V_{2}\rangle +\beta \eta \nabla \left(\nabla \cdot \langle V_{2}\rangle \right)-\nabla \langle p_{2}\rangle =\langle \rho_{1}\partial_{t}V_{1}\rangle +\rho_{0}\langle \left(V_{1}\cdot \nabla \right)V_{1}\rangle$$
in which $\langle V_{2}\rangle$ is the streaming velocity.

### 2.3. Boundary Conditions

#### 2.3.1. Microfluidic Flow

Fully-developed flow and zero pressure were imposed at the inlet and the outlet, respectively. A no-slip boundary condition was considered on all walls as well as at the spheroid and the culture medium interface.

#### 2.3.2. Transport of Dilute Species

At the inlet, constant concentrations of oxygen ($c_{0}{}_{O_{2}}$) and glucose ($c_{0}{}_{Gl}$) inflow were applied. At the outlet, the gradient of concentrations for both glucose and oxygen were set to be zero. All walls are impermeable, and as a result, a no-flux boundary condition was used for them. At the culture medium’s interface with the cell aggregates, concentration jump boundary condition was applied by utilizing a solubility coefficient (*S*) as in Equations (17) and (18) [26]:(17)$$c_{O_{2}.aggregate}=S_{O_{2}-CT\;vs.\;H_{2}O}\times c_{O_{2}.medium}$$
(18)$$c_{Glucose.aggregate}=S_{Glucose-CT\;vs.\;H_{2}O}\times C_{Glucose.medium}$$

Conservation of fluxes was also fixed at the interface of the culture media and the spheroid [26]:(19)$$J_{Glucose.aggregate}=J_{Glucose.medium}$$
(20)$$J_{Oxygen.aggregate=J_{Oxygen.medum}}$$

In addition, concentration conditions that describe the necrotic and the quiescent zones are as follows [48,49]:

For the necrotic zone:(21)$$c_{O_{2}.aggregate<0.002644\;\mathrm{mM}\;\left(equevalent\;to\;2\;\mathrm{mmHg}\;oxygen\;partial\;pressure\right)\;\&\;C_{Glucose.aggregate}<0.2\;\mathrm{mM}}$$

For the quiescent zone:(22)$$c_{O_{2}.aggregate<0.01322\;\mathrm{mM}\;\left(equevalent\;to\;10\;\mathrm{mmHg}\;oxygen\;partial\;pressure\right)\;\&\;C_{Glucose.aggregate}<0.5\;\mathrm{mM}}$$

#### 2.3.3. Acoustic

The first-order velocity boundary condition is as follows:(23a)$$V_{1}=0,\;\;\;\;On\;non-actuated\;walls$$
(23b)$$n\cdot V_{1}=-i\omega d_{0}e^{-i\omega t},\;\;\;\;On\;the\;actuated\;wall$$
(23c)$$\sigma_{1}\cdot n=0,\;\;On\;the\;inlet\;and\;outlet\;boundaries$$

In Equation (20), $d_{0}$ is the amplitude of the wall-normal displacement and $\sigma_{1}$ is the stress tensor which is based on the first-order velocity. Outlet and inlet are both open boundaries that pose no stress on the acoustic waves (no physical barrier and no change of the medium or properties across these boundaries).

For the first-order temperature, all the boundaries are considered to be isothermal, and for the second-order velocity equation, the no-slip boundary condition was applied on the walls, and a fully developed boundary condition was considered for the inflow and outflow, as described in Section 2.3.1.

All constants and parameters of the governing equation and their boundary conditions are presented in Table 2.

*Q* and $d_{0}$ are the main parameters of this study. Their values vary in the range of the values that were used in other similar works. Initial concentrations of glucose and oxygen are both the conventional molarities in the medium containing glucose, e.g., Dulbecco’s Modified Eagle Medium (DMEM). All of the Michaelis-Menten parameters are also the values used in the literature for cancerous cells in spheroids. Regarding the simulation frequency, in microfluidics, it is common to use a frequency in Megahertz (MHz), as the wavelength should be smaller than the minimum feature of the geometry. Here, a frequency of 1 MHz has been chosen whose wavelength is comparable with the geometry’s minimum feature and is considered a generic value. All the references for the parameters can also be found in Table 2.

### 2.4. Numerical Method

The abovementioned equations were discretized and solved utilizing COMSOL Multiphysics version 5.5. All the domains in all the physics have triangular element meshes. A boundary layer mesh was generated for all the wall boundaries to completely capture the acoustic streaming effects taking place within the viscous penetration depth. For the convergence criteria, a residual value of $10^{-6}$ was set for Navier-Stokes and continuity equations, $10^{-3}$ was set for the transport equation, and $10^{-6}$ was used for thermoviscous acoustic equations.

The numerical procedure to solve the present multiphysics problem is as follows: First, thermoviscous equations were solved to find the first-order pressure and velocity. Having obtained the mentioned fields, they are inserted in Equation (15) as source terms to obtain the second-order terms, which are the fluid dynamic velocity and pressure of the system. Finally, the obtained velocity and pressures are used to solve the diluted species transport equations and to find the oxygen and glucose concentrations within the domain.

### 2.5. Mesh-Independent Study

Specific care was given for the mesh generation process pertinent to the thermoviscous and laminar flow solutions [43]. The viscous penetration depth, which is defined by the relation δ=√(2ν/ω)=0.52 nm, was divided into six layers near the wall to fully capture the acoustic streaming phenomenon. In the well and the free stream domain, maximum element size was set to 10 and 20 times the viscous penetration depth, respectively.

According to Muller et al. [43], streaming velocity was the last quantity to converge. As a result, in this study, a grid independence study is conducted on the shear stress, which is derivative of velocity and more sensitive to velocity variations and a better indicator of mesh independence. Accordingly, a grid study on shear stress aims for not only a thermoviscous acoustic solution but also laminar flow. To do so, the maximum value of fluid shear stress (FSS) on the spheroid was considered against mesh refinement. On the upper half of the spheroid, the flow of the fluid in the perfusion channel is mainly responsible for the FSS, and for the lower half, the acoustic field is responsible. Another point of this consideration is to discover if the magnitude of FSS after acoustic application does not surpass the allowable range mentioned in the literature [52]. In Figure 2, details of this investigation are presented.

The mesh study’s error remains below three percent in this section. To prove that the mass transport solution is not dependent on generated meshes, the average magnitude of oxygen’s concentration in the whole spheroid was measured while the number of mesh elements increased. Simulation outcomes proved the point that the solution remains constant, independent of the number of meshes. Figure 2 shows the convergence of grid study. With 40,368 elements, the error will be less than one percent compared to the finer sizes of meshes.

### 2.6. Validation of the Study

A step-by-step approach to the validation was implemented. As described in Section 2.2, the model is a multiphysics problem in which laminar flow, mass transport, and thermoviscous acoustic equations are involved. The first step is to show that laminar flow and mass transport are modeled correctly in our work. We have previously validated the accuracy of our proposed numerical model for modeling laminar flow and mass transport in similar spheroid-on-chip systems [22,26]. The present model is an extension to those previously published articles. The difference is that here, a piezo transducer is integrated to the system as a potential solution to overcome the challenges regarding necrotic and quiescent zones formation inside a spheroid.

The second step is to prove that we are able to solve acoustic equations in 2D domains correctly in COMSOL. For the validation of the acoustic model, the model of Muller et al. [43] was regenerated and simulated, whose geometry (a simple 2D rectangular-shaped domain) is shown in Figure 3.

The number given to each boundary in Figure 3 specifies its boundary condition defined as below:(24a)$$①\{\begin{array}{c} \mathrm{First}-\mathrm{order}\;\mathrm{Fields}:\;\{\begin{array}{c} T_{1}=T_{0},\\ n\cdot V_{1}=\omega d_{0}e^{-i\omega t}\; \end{array}\\ \mathrm{Second}-\mathrm{order}\;\mathrm{Fields}:\{\begin{array}{c} V_{2}=0,\\ \int^{\;}PdA=0\;\left(in\;the\;domain\right) \end{array} \end{array}$$
(24b)$$②\{\begin{array}{c} \mathrm{First}-\mathrm{order}\;\mathrm{Fields}:\;\{\begin{array}{c} T_{1}=T_{0},\\ V_{1}=0\; \end{array}\\ \mathrm{Second}-\mathrm{order}\;\mathrm{Fields}:\;\{\begin{array}{c} V_{2}=0,\\ \int^{\;}PdA=0\;\;\left(in\;the\;domain\right) \end{array} \end{array}$$

It should be noted that here, instead of fixing the pressure in a point for the second-order equations, the average of the pressure in the whole domain has been set equal to zero. The geometrical and simulation parameters of this problem are presented in Table 3.

In Figure 4a–c, vertical velocity, near-wall vertical velocity, and horizontal velocity are respectively compared between this study and [43], all of which are plotted on the white dashed line located W/4 to the right of the origin in Figure 3. From Figure 4, it is evident that the results are compatible which gives rise to the validity of the acoustics part of this study. Because of the small magnitude of the horizontal velocity (in the order of mm/s) compared to the vertical one (in the order of m/s), some fluctuations are seen in Figure 4c. In Figure 4d, first-order pressure inside the computational domain is shown. This sub-figure is also identical to Figure 4a of [43], which was obtained with the boundary conditions given in Equation (24) and parameters given in Table 3.

## 3. Results and Discussion

### 3.1. Conventional Spheroid-on-Chip Platform (No Acoustic)

In this section, the model was simulated without acoustic integration to obtain a broad view of the physics of the problem. Figure 5a shows velocity contour and streamlines inside the conventional microfluidic system (without acoustic). Streamlines are almost parallel and the fluid flows from the inlet to the outlet almost straightly. The flow inside the microwell and around the spheroid is negligible. In Figure 5b,c, oxygen and glucose distributions and their fluxes are observable, respectively. Diffusion is mainly responsible for transporting nutritious species to the cell aggregate, while the convection term in the well and adjacent to the spheroid is not strong enough.

### 3.2. Acoustic Spheroid-on-Chip Platform

After introducing an acoustic field ($d_{0}=0.5\;\mathrm{nm}\;\mathrm{and}\;f_{0}=1\;\mathrm{MHz}$) to the system, for the same flow rate of 1 μL/min, flow’s streamlines start to form some vortexes near the microwell, which enhances convention inside the microwell (Figure 6a). Oxygen and glucose concentration distributions and their fluxes also are presented in Figure 6b,c, respectively. The results show that fluxes of nutrition towards the cell aggregate are enhanced in the presence of an acoustic field.

In the following subsections, the effect of two principal factors, namely boundary displacement amplitude and the inlet flow rate, on the flow pattern inside the chip and the concentration level of glucose and oxygen in the vicinity of the spheroid will be studied.

### 3.3. Boundary Displacement Amplitude

The effect of boundary displacement amplitude ($d_{0}$), which is controlled by the properties of the piezoelectric substrate and the actuation power, on the spheroid culturing is studied in this section. $d_{0}$ represents the displacement amplitude of the actuated boundary and in practical setups is usually measured to be in the range of 0.1–0.5 nm [53]. In this section, frequency and flow rate are both kept constant at 1 MHz and 1 μL/min, respectively. Figure 7a shows the effect of $d_{0}$ on glucose and oxygen concentrations within the spheroid. As the amplitude of boundary vibration increases, the acoustic streaming effects escalate which in turn influences and disturbs the fluid flow pattern more intensely, and consequently the convection in the microwell around the spheroid is improved. In the absence of any disturbing factor, the concentration gradient increases in the fluid because of the presence of a multicellular spheroid (which acts as a nutrient sink). However, with the presence of a disturbing factor such as a penetrating ultrasonic wave applied through wall fluctuations, the flow mixes and the nutrient gradient in the vicinity of the spheroid decreases, which brings about a more homogenous culture medium in the spheroid proximity. The higher the amplitude of the wall displacement amplitude, the higher the level of mixture in the flow, and the more homogenized the flow. Therefore, the mass transfer in the vicinity of the spheroid will be improved, and oxygen and glucose can more easily and effectively diffuse to the core of the cell aggregate. Interestingly, for oxygen, increasing $d_{0}\;$ from 0.2 nm to 0.5 nm leads to a 228% rise in the average concentration of oxygen in the spheroid. For glucose, although the amplification is not massive, employing acoustics nonetheless improves the distribution of glucose in the spheroid. The improvement of oxygen/glucose diffusion into the core of the spheroid ends in the shrinkage of both necrotic and quiescent zones and consequently the growth of the proliferation zone.

While introducing acoustics to the chip can vividly enhance the nutrient-enrichment of the spheroid, it is important to make sure that the integration of the acoustics is not damaging cells by imposing a high magnitude of shear stress on them or inhibiting the spheroid entrapment in the microwell due to exerting lift force on it. From the hydrodynamic point of view [54], the maximum allowable fluid shear stress on the cells is $0.5\;\mathrm{dyne}/{\mathrm{cm}}^{2}$. In addition, the extra produced lift force due to the acoustic field should not exceed the downward force which is the net of weight and the spheroid’s buoyance force. Figure 7b proves that from the hydrodynamic point of view $d_{0}\;$is within the safe range and acoustics neither hamper the spheroid culturing process nor adversely affect cell viability due to shear stress. Figure 7c illustrates how increasing $d_{0}$ eliminates unwanted quiescent/necrotic zones and causes the proliferating zone for the oxygen to grow inside the spheroid.

### 3.4. Flow Rate

Flow rate is the main controllable parameter that can directly govern the hydrodynamic field inside microchannels. Higher flow rates correspond with more mass of the species in the computational domain and consequently higher average concentration of oxygen/glucose inside the spheroid. Figure 8a shows how flow rate affects concentration and compares the average glucose/oxygen concentrations in acoustic ($d_{0}$ = 0.5 nm) and non-acoustic systems as a function of flow rate. As expected, there is an upward trend between the flow rate and the average concentration. For oxygen, for example, a 300% increase in the flow rate from 1 μL/min to 4 μL/min leads to the rise of average concentration from 0.013 mM to 0.028 mM. However, by keeping the flow rate at 1 μL/min and only introducing the acoustic field to the domain, oxygen’s average concentration will be pumped up to 0.046 mM.

Since acoustic waves inside the microwell enhance the hydrodynamic flow around the sphere, it might be interesting to also compare lift force and maximum shear stress in both acoustic ($d_{0}$ = 0.5 nm) and non-acoustic ($d_{0}\;$ = 0 nm) modes and also investigate how flow rate affects it. As shown in Figure 8b, it is true that the addition of acoustics to the system exerts more shear stress and lift force on the cell aggregate, but it is well below the thresholds.

Figure 8c demonstrates necrotic and quiescent zones’ shrinkage as the flow rate is increased. When the acoustic field was introduced to the system, the proliferating zone’s share from the spheroid rose, and it can also be seen that with acoustics ($d_{0}$ = 0.5 nm), the downward trend in the shrinkage of the necrotic zone was also sharper against the increase in the flow rate. In Figure 7c, it is good to compare the first (with acoustics, $d_{0}\;$= 0.5 nm, *Q* = 1 μL/min) with the last (no acoustics, q = 12 μL/min) graphical illustrations. Based on simulation results, a low flow rate of 1 μL/min with an acoustic boundary displacement of 0.5 nm results in 66.8% of the proliferation zone, 19.9% of the quiescent zone, and 13.3% of necrotic zone. To achieve a similar percentage of zones without acoustic waves, the flow rate should be increased 12 times. This proves that acoustics improves cell viability astonishingly on the chip without the need to consume more reagents. About 100% increase was observed in the area of proliferating zone after applying the acoustic field at each flow rate.

Based on the results provided in this work, oxygen is the critical element in the necrotic zone formation since it is depleted faster than glucose. In all cases of this study, the concentration of glucose within the spheroid was more than the critical value for necrosis (Equation (22)). As discussed in the introduction part of this work, necrosis can occur due to the lack of oxygen, glucose, or both. The Michaelis-Menten parameters, specifically $K_{m}$, control the consumption of species and consequently necrotic/quiescent zones’ formation. Additionally, other parameters such as geometry, flowrate, initial concentration, and diameter of the spheroid can also play an important role in this regard. In this paper, owing to the geometry and the values used for the parameters, glucose is not the main player in the necrosis. However, to show the importance of glucose in such studies and also to discuss how lack of glucose and oxygen independently contribute to the shrinkage of the proliferating zone, we simulated the model again with the parameters used in Figure 8c for Q=1 μL/min, except that this time, the initial concentration of glucose was reduced to 1 mM (low-glucose culture media). With this new condition, glucose also caused necrosis within the spheroid. The middle section of Figure 9 illustrates the formation of necrotic and quiescent zones due to the shortage of glucose. Without acoustics, lack of glucose is also responsible for forming the necrotic zone (22%), although its impact is still less than oxygen (45%). Regarding the formation of the quiescent zone without acoustics, the impact of glucose (66%) is larger than that of oxygen (22.3%). This implies that glucose is the critical element in forming the quiescent zone, leading to the shrinkage of the proliferating zone. With the introduction of acoustics to the system, distributions of oxygen and glucose are significantly enhanced and, consequently, the share of the proliferating zone increases from 11.2% to 54.9%. These results show that oxygen and glucose can both be the cause of necrosis and the formation of the quiescent zone, and the integration of acoustics to the system improves the conditions for both factors.

It should be noted that we only studied the model in a 2D domain and considered only the effect of mass transport and flow on necrosis. Several other phenomena can cause necrosis in spheroids, which are beyond the scope of this paper, but they are crucial to be considered in experimental studies and design considerations. Surface contact with the wall, for example, is also a crucial point that causes necrosis in 3D spheroids. As also confirmed in our previous studies [22,24], the spheroid’s necrotic region increases when there have been contacts between the spheroid’s cells and the surface, or in situations where the distance between cells and the surface is small. The reason is that cells in these regions do not have access to nutrition, and diffusion is hampered. In the present study, we observed that the necrosis core is not in the spheroid’s centre. Indeed, it forms closer to the bottom part, which is near to the surface. Homotypic cell-cell contact also is another example. It triggers cell death and the formation of necrosis zones in 3D spheroids [55].

## 4. Conclusions

Using numerical simulation, we conducted a proof-of-concept of implementing acoustic spheroid-on-chip microfluidic platforms to increase culturing efficiency. The goal was to engineering a microsystem to evade the pitfalls of conventional spheroid-on-chip kits. By implementing the proposed approach, a better cell viability rate is achievable without the need for utilizing complex geometries, increasing the flow rate, or even culturing spheroids with smaller radii.

First, the proposed geometry and the concept of the model were described. Next, the numerical method and governing equations were presented in detail. Finally, simulation results were reported and discussed. It was observed that by implementing acoustic waves, the pattern of the flow, and the distribution of concentration and mass fluxes throughout the microfluidic chip were changed. As a result of the acoustic streaming phenomenon, fluidic convection was enhanced in the microwell and around the spheroid. Consequently, more oxygen/glucose was transported to the vicinity of the spheroid, which itself ended up in higher concentrations of nutritious species inside the cell aggregate.

The effect of acoustic power on spheroid culturing was studied via the wall displacement amplitude ($d_{0}$). It was shown that increasing this parameter results in the extension of the proliferating zone. Compared to the non-acoustic mode, with a constant, small flow rate, the share of proliferating zone in the spheroid can be doubled with the aid of acoustics. The effect of flow rate was also investigated. It was concluded that although higher flow rates cause shrinkage in both necrotic and quiescent zones, the introduction of an acoustic field to the system enhances this effect without the need for consuming more culture media or treating reagents.

The possible drawback of acoustic integration is the production of high magnitudes of fluid shear stresses and lift forces. We numerically calculated both shear stress and lift forces and proved that they were in the safe range with no adverse effects on the cell aggregate. In summary, the proposed acoustic microfluidic cell-culturing model has the potential to open up new avenues for in vitro cell culture platforms, especially for cancer drug screening on 3D spheroids and other related biological assays.

## Figures and Tables

**Figure 1 sensors-21-05529-f001:**
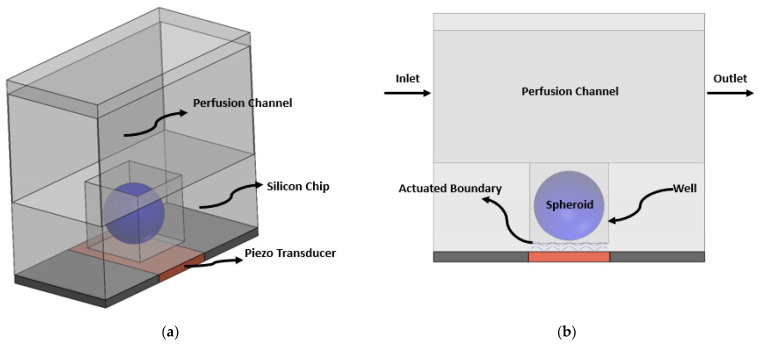
(**a**) 3D format and (**b**) 2D format of the schematic of the proposed acoustic-microfluidic system for 3D cell culture The geometry of a spheroid by its nature is 3D (i.e., a sphere). When modeling it in a 2D domain as a circle, we are only observing the midplane of the system and neglecting the effect of the third dimension on the system. Theoretically, this 2D model represents more quantitatively an infinite cylindrical cell aggregate and more qualitatively a spherical cell aggregate. Using a 3D model was computationally prohibitive in this work, as the number of mesh elements increases non-linearly with the frequency, and for our MHz-range simulation, a 3D solution was not affordable. While the simulation results are not exactly the same for a 3D model, they provide a good qualitative solution for the proof-of-concept study. Additionally, we hold the conviction that our simulation results are much more comparable to a spherical cell aggregate model rather than a finite cylindrical one. Consider a finite cylinder. For this model, there is a sharp difference in the flow field between the middle of the domain and its ends. However, there is no such difference between the middle and the end sections for a spherical model. The flow cell consists of a perfusion channel and microwell in which the spheroid is cultured. Beneath the flow cell, a piezo transducer is located.

**Figure 2 sensors-21-05529-f002:**
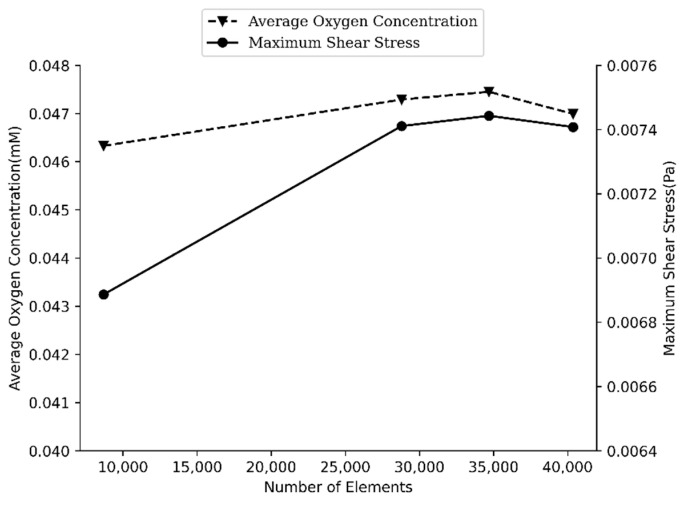
Mesh study graph for laminar flow and mass transport equations. As the grid size reduced in a step-by-step manner, both maximum shear stress and average oxygen concentration tend to be constant and independent of the number of the grids.

**Figure 3 sensors-21-05529-f003:**
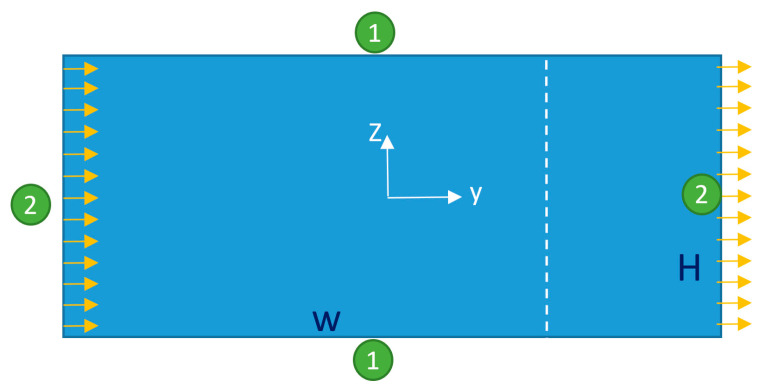
The geometry of the model simulated in [43], regenerated here for the sake of validation. The orange arrows represent the actuated velocity. The number adjacent to each boundary specifies its boundary condition which is described in Equation (24) with the same boundary numbers. The line graphs of Figure 4 are plotted on the white dashed line shown in this figure.

**Figure 4 sensors-21-05529-f004:**
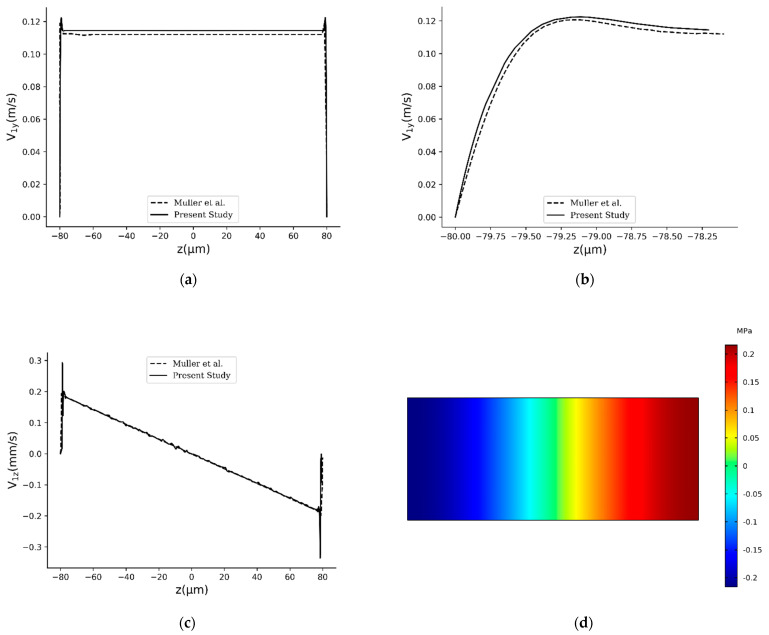
Comparison between the results obtained in this study for the geometry and boundary conditions of the problem in [43] and present results. (**a**) Comparison between the vertical velocities, (**b**) comparison between the vertical velocities in the near-wall region, and (**c**) comparison between the horizontal velocities. All the sub-figures (**a**–**c**) are plotted on the white dashed line shown in Figure 3. Sub-figure (**d**) depicts the pressure contour for the problem, which is identical to its counterpart in [43] (Figure 4a in that study).

**Figure 5 sensors-21-05529-f005:**
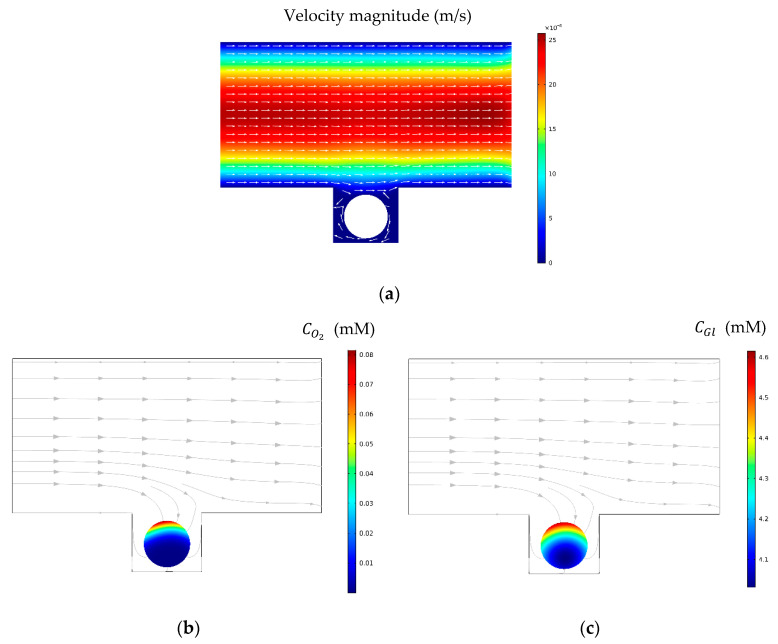
(**a**) Velocity contour and streamlines without acoustics. Velocity distribution is approximately similar to the laminar flow pattern inside a rectangular domain. (**b**,**c**) Oxygen and glucose concentration distribution and their fluxes. Here, the flow rate was set to be 1 μL/min.

**Figure 6 sensors-21-05529-f006:**
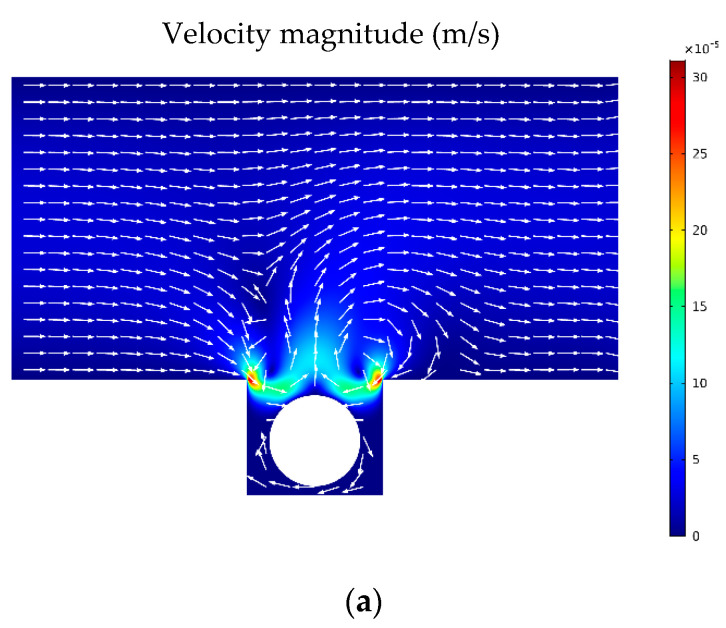
(**a**) Streamlines in the presence of the acoustic field. Near the microwell, the flow is mostly affected and forms a pattern similar to a vortex. This also leads to convection enhancement inside the well and around the spheroid. (**b**,**c**) Oxygen and glucose distribution and their fluxes, respectively, when the acoustic field is present. Acoustic field improved the proliferation zone significantly inside the spheroid.

**Figure 7 sensors-21-05529-f007:**
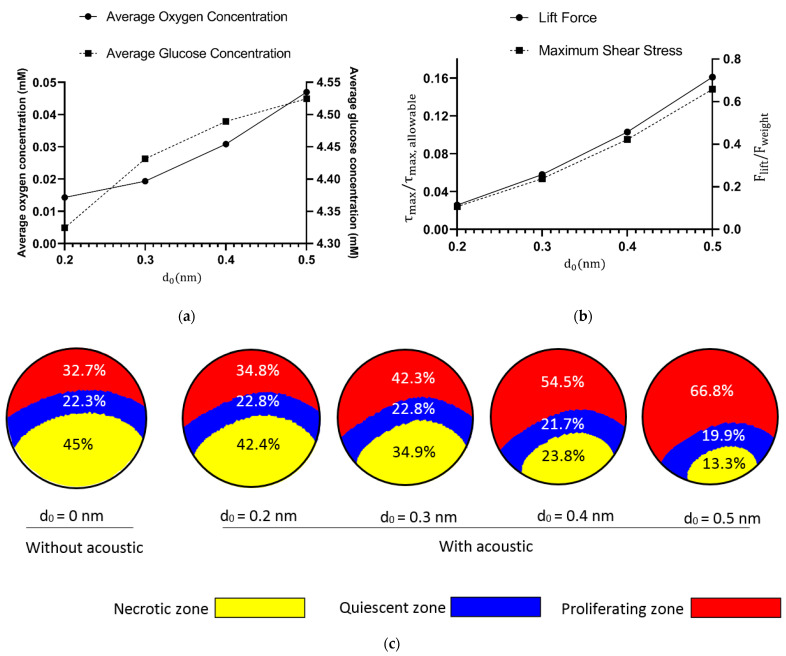
(**a**) Effect of $d_{0}$ on glucose and oxygen concentrations inside the spheroid. With an increase in the magnitude of $d_{0}$, more oxygen and glucose will be found inside the cell aggregate, and the better cells in the inner side and the core will be nourished. (**b**) While increasing $d_{0}\;$ is beneficial due to enhancement of oxygen and glucose concentrations, it is not also causing any side effects such as high magnitudes of fluid shear stresses or lift forces. Here, $\tau_{max}\;$ represents the maximum value of shear stress which is exerted to the peripheral boundary of the spheroid. (**c**) Graphical illustration of oxygen’s proliferation zone after applying the acoustic field. Before applying acoustic to the microchip, the proliferation zone had only a 32.7% share of the spheroid. With the acoustic field, this share is increased to 66.8%.

**Figure 8 sensors-21-05529-f008:**
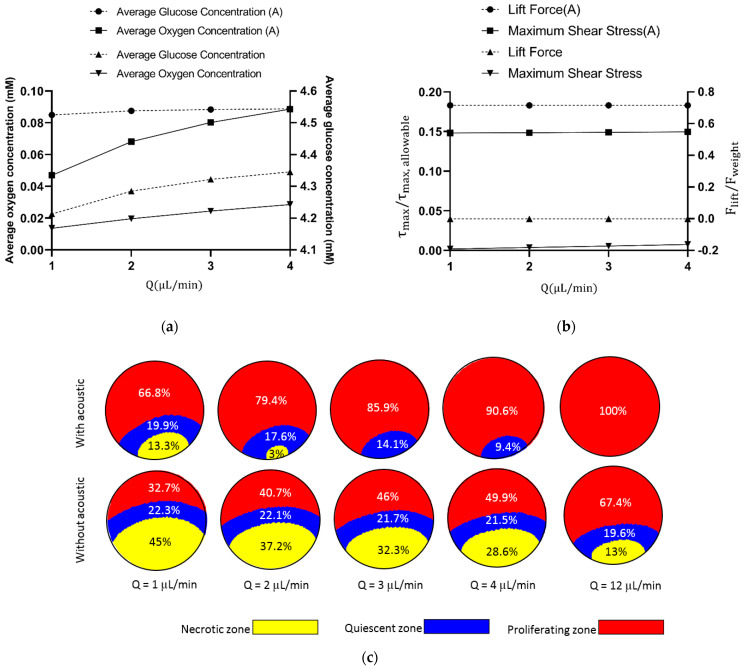
(**a**) Effect of flow rate on oxygen/glucose concentration with/without acoustics. As shown in the figure, the acoustic field enhanced concentrations greatly without the need for any increases in the flow rate. (**b**) A study of the effect of flow rate and acoustic integration to the system on maximum fluid shear stress and the lift force. The acoustic wave is amplifying these two parameters, but they are still in the safe range for this application. (**c**) A comparison in necrotic/quiescent shrinkage with a flow rate between acoustic and non-acoustic modes. Increasing the flow rate helps the growth of the proliferation zone. The acoustic field solely can perform better without the need for any increases in the flow rate. Approximately, acoustics improves the proliferating zone by 100% at each flow rate.

**Figure 9 sensors-21-05529-f009:**
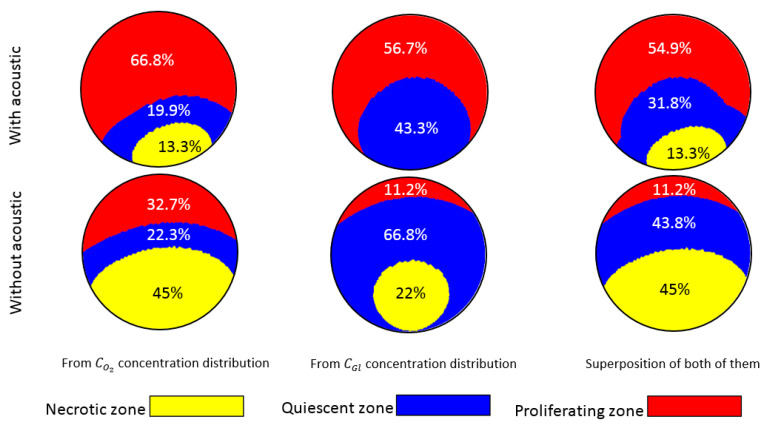
Distribution of glucose and oxygen with and without acoustics. This figure aims to show how oxygen (left column) and glucose (middle column) solely contribute to the formation of necrotic and quiescent zones. Without acoustics, the shortage of oxygen leads to a 32.7% proliferation zone (bottom-left), while this is only 11.2% for glucose (bottom-center). The huge loss of the proliferation zone is mainly due to the large quiescent zone (66.8%) caused by the lack of glucose. The combinatorial effect is 11.2% of proliferating zone, 43.8% quiescent zone, and 45% necrotic zone (bottom-right). With acoustics, the necrotic zone from glucose shortage is omitted completely and is decreased to 13.3% from the lack of oxygen. The quiescent zones of oxygen and glucose are also reduced in size, and consequently, a noticeable improvement in the combinatorial share of the proliferating zone is observed (54.9%, top-right).

**Table 1 sensors-21-05529-t001:** Values of the geometrical parameters used in computational modeling.

Parameter	Value
Spheroid diameter	300 μm
Well height	380 μm
Well width	450 μm
Channel height	1000 μm
Channel length	2000 μm

**Table 2 sensors-21-05529-t002:** Detailed values and descriptions of all the parameters used in the governing equations used in this study.

Parameters	Descriptions	Values	References
Q	Inflow	1–12 μL/min	[26]
$c_{0}{}_{O_{2}}\;$	Inlet concentration of oxygen	0.2 mM	[22]
$c_{0}{}_{Gl}$	Inlet concentration of glucose	5 mM	[50]
$D_{O_{2}-H_{2}O}$	Diffusion coefficient of oxygen through H_2_O	$2.6\times 10^{-9}{\;m}^{2}/s$	[22]
$D_{O_{2}-Sph}$	Diffusion coefficient of oxygen through the cell aggregate	$1.83\times 10^{-9}{\;m}^{2}/s$	[22]
$D_{Gl-H_{2}O\;}$	Diffusion coefficient of Glucose through H_2_O	$9.27\times 10^{-10}\;m^{2}/s$	[22]
$D_{Gl-Sph\;}$	Diffusion coefficient of glucose through cell aggregate	$2.7\times 10^{-10}{\;m}^{2}/s$	[22]
$S_{O_{2}-Sph\;vs.\;H_{2}O}$	Solubility coefficient of oxygen in the cell aggregate vs. H_2_O	4.81	[22]
$S_{Gl-Sph\;vs.\;H_{2}O}$	Solubility coefficient of glucose in the cell aggregate vs. H_2_O	1	[22]
$V_{max}{}_{O_{2}}$	Maximum reaction rate of oxygen	0.0203 mM/s	[22]
$V_{max}{}_{Gl}$	Maximum reaction rate of glucose	0.01076 mM/s	[22]
$K_{m}{}_{O_{2}}$	Michaelis-Menten constant of oxygen	0.00463 mM	[22]
$K_{m}{}_{Gl}$	Michaelis-Menten constant of glucose	0.04 mM	[22]
$f_{0}$	Actuation frequency	1 MHz	-
ρ	Fluid density	$993.3\;\mathrm{kg}/m^{3}$	[22]
μ	Fluid dynamic viscosity	$6.92\times 10^{-4}\;\mathrm{Pa}\cdot s$	[22]
$\mu_{B}$	Fluid bulk viscosity	0.0024 Pa·s	[43]
Cp	Fluid specific heat at constant pressure	$4.18\frac{\mathrm{kJ}}{\mathrm{kg}\cdot K}$	[43]
$\alpha_{0}$	Fluid thermal expansion	$2.75\times 10^{-4}\;1/K$	[43]
$\beta_{0}$	Fluid isentropic compressibility	$4.48\times 10^{-10}\;1/\mathrm{Pa}$	[43]
$d_{0}$	Wall displacement amplitude (Equation (21))	0.1−0.5 nm	[43,51]
C	Sound velocity in the fluid	1502 m/s	[43]

**Table 3 sensors-21-05529-t003:** Parameters of the model used in Ref. [43], which have been re-simulated here.

Parameter	Value
W	380 μm
H	160 μm
f	1.97 MHz
$T_{0}$	25 °C
$d_{0}$	0.1 nm

## Data Availability

The datasets used and/or analyzed during the current study are available from the corresponding author on reasonable request.

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
