# Peer review of "A Proof-of-Concept Study Using Numerical Simulations of an Acoustic Spheroid-on-a-Chip Platform for Improving 3D Cell Culture"

_sensors, 2021, doi:10.3390/s21165529_

Round 1
Reviewer 1 Report
This paper describes a numerical simulation of oxygen and glucose transportation into a spheroid cultured in a microfluidic channel. By applying acoustic waves, the transportations of oxygen and glucose around the spheroid were enhanced, resulting in that the proliferating zone in the spheroid increased. This result indicates that the acoustic microfluidics systems would provide long-term spheroid culturing without the need for consuming more culture media.
The experimental conditions and procedures for simulations are well written, and the simulation result would widely attract attention from various research fields, including microfluidics, bioengineering, drug discovery, and regenerative medicine. However, the authors should revise the results and discussion section to be clear about the results. There are several concerning points as listed below.
- In table 2, the authors should mention which values have been used from references [21, 33, 40].
- In “2.6 Validation of the study”, the authors should be clear about the simulation model (Figure 3) and the simulation result (Figure 4). For example, how did the authors apply acoustic waves? How acoustic waves (frequency and displacement) did authors apply in the model? In figure 4, the authors should show not only the vertical velocity but also horizontal velocity, color plots of pressure, vertical and horizontal velocities.
- In Figures 5 and 6, the authors should add the unit to the simulation results.
- Although the authors stated that “Interestingly, for oxygen, increasing… (Line: 360-363)”, why did the oxygen concentration increase with increasing d0? Since it is the critical factor for spheroid culturing, the authors should discuss it in detail.
- The authors defined the necrotic and quiescent zones as “??????????? <0.002644 ?? (?????????? ?? 2 ???? ?????? ??????? ????????) & ????????.????????? < 0.2 ??” and “??2.????????? < 0.01322 ?? (?????????? ?? 10 ???? ?????? ??????? ????????) & ????????.????????? < 0.5 ??” at the line 228. However, the glucose concentration in the spheroid was much higher than the threshold in any cases (w/ and w/o acoustic waves). Thus, the reviewer guesses that the necrosis only relates to the oxygen concentration in the microfluidic platform. If it is correct, what is the condition that glucose is not sufficient? Also, does it really occur?
- In figure 8 b, is d0 0.5 nm? The values of maximum shear stress and lift force are different from these values in figure 7 b. The authors should check it and mention the d0 value.
- What kind of simulation software did the authors use?
- Finally, the title is unsuitable for this article because the authors did not fabricate microfluidic chips and culture spheroids. Also, the authors did not discuss the long-term influence of applying acoustic waves for spheroids.
Reviewer 2 Report
This paper proposes, theoretically and supported by simulations, a microfluidic device with an embedded piezo transducer that provides an acoustic stimulus that, presumably, results in a decrease in necrosis of 3D cultures in a spheroid. The paper is based on simulations. Although the models used for laminar flow (LM), mass transport (MT) and acoustics have been previously reported, the paper presents a translation on how the flow and transport of diluted species (i.e. analytes) could result in an improvement of spheroids. The term "acoustics spheroid-on-a-chip" is proposed, however the approach is similar to previously reported literature, including Surface Acoustic Waves and 3D tissues (spheroids) and others (e.g. https://doi.org/10.1088/1361-6528/aae4f1). Although I consider the paper is not ready for publication in Sensors in its current state, I believe the paper could be eventually considered, so I recommend they consider resubmission after appropriate modifications.
Comments:
- Although it seems like an interesting approach, the paper relies only on finite element analysis (FEA) and provides no experimental results.
- The study is presented at a proof-of-concept level, and the findings are based on the change in flow patterns and transport of analytes due to the acoustic stimulus.
- The title is misleading as it could lead to the readership to think it is an actual experimental study. The title must be adapted to communicate the right message.
- Similar to the title, the abstract contains several misleading messages or sentences. E.g. "We show that such an approach enhances cell viability and shrinks necrotic and hypoxic zones in these spheroids...". This statement is too bold and the authors should consider lowering the tone and conveying the right message. I.e. that the study suggests that the transport of anlaytes is enhanced due to the acoustic stimuli that, in vitro/vivo, could result in..."
- The paper has several problems related to terminology. E.g. in the Abstract: "... and finite element numerical simulation are discussed in detail." Do they mean Finite Element method? Finite Element Analysis? This problem is found several times, with different terms throughout the manuscript. A revision and more careful use of the terminology is strongly recommended.
- Table 2 presents the values used in the simulations, but there is no concise justification of the values used.
- In Section 2.6, they mention that they have previously validated the accuracy of the proposed numerical model for LF and MT, and they are using the model by Muller et al for the acoustic model. Then, they should clarify the novelty of the presented study in terms of FEA. It is understood that the translation to spheroid necrosis is a contribution, novelty in the context of analysis must be clarified as well.
- The results presented include only the effects of the flow and mass transport. However, in real-world 3D tissue applications, surface contact is one of the main reasons for spheroids to necrotize. This aspect should be definitely discussed in the paper.
Round 2
Reviewer 2 Report
The authors have made a major revision of the manuscript and changed the focus of it so it conveys the right message - a FEM-based study of an acoustic spheroid-on-a-chip platform that could lead to improve 3D cell cultures, mostly due to the reduction in necrosis in the spheroid, which is a common problem in microfluidic 3D cell cultures, as it has been widely published in the literature (i.e. droplet-based methods). With the major revision, the manuscript could be considered for publication if the editor considers is suitable in current form.
Author Response
We would like to sincerely thank the reviewer for his/her insightful comments. We are so happy that he/she finds suitable for publication.